# Semantic regularization of electromagnetic inverse problems

Hongrui Zhang[1,5], Yanjin Chen[1,5], Zhuo Wang[1], Tie Jun Cui ⓘ [2,3] ✉,
Philipp del Hougne ⓘ [4] ✉ & Lianlin Li ⓘ [1,3] ✉

Solving ill-posed inverse problems typically requires regularization based on prior knowledge. To date, only prior knowledge that is formulated mathematically (e.g., sparsity of the unknown) or implicitly learned from quantitative data can be used for regularization. Thereby, semantically formulated prior knowledge derived from human reasoning and recognition is excluded. Here, we introduce and demonstrate the concept of semantic regularization based on a pre-trained large language model to overcome this vexing limitation. We study the approach, first, numerically in a prototypical 2D inverse scattering problem, and, second, experimentally in 3D and 4D compressive microwave imaging problems based on programmable metasurfaces. We highlight that semantic regularization enables new forms of highly-sought privacy protection for applications like smart homes, touchless human-machine interaction and security screening: selected subjects in the scene can be concealed, or their actions and postures can be altered in the reconstruction by manipulating the semantic prior with suitable language-based control commands.

Inverse problems, omnipresent in most areas of science and engineering, are notoriously difficult to solve due to their ill-posed nature[1–7]. Generally speaking, an inverse problem seeks to find the "cause" that gave rise to an observed (or desired) "effect"[8]. For instance, wave-based imaging seeks to reconstruct the material properties of a scene based on observations of how the scene scatters known impinging waves[9]. The ill-posedness of an inverse problem originates from the low-dimensional and noisy nature of the available measurements: multiple distinct causes can plausibly explain the observed measurements. To solve the inverse problem, prior knowledge about the sought-after cause must be introduced to supplement the insufficient measurements. The construction of a modified, approximately well-posed version of the originally ill-posed inverse problem based on prior knowledge is known as regularization. The regularization process can be understood in light of Bayes' theorem[10,11]: our prior knowledge is updated with the new information from the measurements. Pioneered by Tikhonov[12], a wide range of

regularization techniques[2,12–14] has been explored that mathematically formulate their prior knowledge about the unknown. Examples of prior knowledge about the unknown include the fact that it is smooth[15], piece-wise smooth[16], or sparse[17], or that it has a tree-like pattern[18]. More recently, alternative data-driven regularization methods implicitly learned from quantitative calibration data emerged[19–22]. Despite the huge success of both mathematically-formulated and quantitative-data-driven regularizers in mitigating the ill-posedness of inverse problems, these methods struggle or fail to handle prior information originating from human recognition or reasoning because the latter priors are typically formulated semantically rather than mathematically or quantitatively.

Indeed, human natural language is an indispensable means of characterizing, understanding, and reasoning about the world around us and the phenomena we observe as humans. Complex reasoning can be formulated semantically with human natural language, but it would be difficult to convert it to mathematical language. Recent years have

[1]State Key Laboratory of Advanced Optical Communication Systems and Networks, School of Electronics, Peking University, Beijing 100871, China. [2]State Key Laboratory of Millimeter Waves, Southeast University, Nanjing 210096, China. [3]Pazhou Laboratory (Huangpu), Guangzhou, Guangdong 510555, China. [4]Univ Rennes, CNRS, IETR - UMR 6164, F-35000 Rennes, France. [5]These authors contributed equally: Hongrui Zhang, Yanjin Chen. ✉e-mail: tjcui@seu.edu.cn; philipp.del-hougne@univ-rennes1.fr; lianlin.li@pku.edu.cn

witnessed a revolution in natural language processing driven by large language models (LLMs) trained on vast web-scale datasets. This revolution has not only transformed natural language processing itself[23–29] but has also stimulated unprecedented interests in other reasoning-related domains, such as robotics[30–33], computer vision[34–36], code writing[37,38], or material design[39,40]. Interestingly, modern LLMs have powerful zero-shot generalization capabilities[25–27,41–44] and can align text with other modalities (e.g., images and voices). LLMs embed closely related concepts near to each other, i.e., they map distinct texts about the same concept to similar low-dimensional real-valued vectors; for instance, the embedding of the word "peach" is closer to that of "fruit" than to that of "knife". Therefore, LLMs work surprisingly well for unseen samples without any fine tuning[41–44]. Of course, as shown by empirical trends, the zero-shot generalization improves with the model scale, dataset size, and the computational resource dedicated to training[44]. These capabilities of modern LLMs allow us to envision the LLM as powerful tool for capturing and representing human knowledge in order to serve as efficient regularization tool for ill-posed inverse problems when priors are formulated semantically.

In this paper, we propose and demonstrate the concept of LLM-based semantic regularization of ill-posed electromagnetic (EM) inverse problems. We tackle the resulting multi-modal inverse problem with an encoder-decoder deep neural network architecture that we train with the so-called contrastive language-image pre-training method[44]. The encoder embeds the measurements in two outputs, of which one is a semantic embedding that describes the scene in human natural language and carries high-level information about the unknown scene. Based on the two embeddings, the decoder reconstructs the scene in remarkable detail without infringing the visual privacy of subjects in the scene. Importantly, the semantic embedding enables new types of security and privacy-preservation for smart home appliances that require some degree of indoor surveillance. For instance, suppose that encoder and decoder are integrated at the transmitter and receiver, respectively; then, by redacting the semantic embedding before transmission, the sensed information about the subjects is safely encoded and can only be extracted by the intended appliance's receiver. Other possibilities to enhance the privacy-preservation that we explore include the purposeful modification of the semantic embedding with language-based control commands that conceal or alter the appearance of subjects in the reconstruction. Thereby, if, for instance, a smart appliance only needs to monitor an elderly person, the privacy of other inhabitants can be fully protected by concealing them from the reconstructions or by adjusting their posture/action in the reconstructions into a default one that does not reveal their true action/posture. We demonstrate the feasibility of semantic regularization for two important ill-posed EM inverse problems. First, we numerically study a prototypical 2D inverse scattering problem. Second, we experimentally consider compressive programmable-metasurface-based microwave imaging in 3D and 4D (where the fourth dimension represents time) that is envisioned to be a key enabler of smart home appliances. We faithfully expect that our proposed semantic regularization method provides fundamentally new perspectives on ill-posed inverse problems and relevant applications in communications, imaging and beyond.

## Results
### Problem statement
We seek to retrieve a high-dimensional unknown $\mathbf{x} \in \mathbb{R}^N$ from noisy low-dimensional measurements $\mathbf{y} \in \mathbb{R}^n$ which are related to the unknown $\mathbf{x}$ through a mapping operator f, i.e., $\mathbf{y} = f(\mathbf{x}) + \boldsymbol{\epsilon}$, where $\boldsymbol{\epsilon} \in \mathbb{R}^n$ accounts for noise, modeling error and other possible uncertainties. This inverse problem is ill-posed due to the lack of a unique solution: an infinite number of solutions $\hat{\mathbf{x}}$ can "explain" the measurements $\mathbf{y}$ well but most of these solutions are not meaningful. Hence, regularization methods are necessary which modify the

problem and introduce prior knowledge to overcome the ill-posedness. In particular, the solution space of $\mathbf{x}$ can be narrowed down to an $m$-dimensional ($m < n$) manifold $S$ via a transforming operator d that maps a solution $\hat{\boldsymbol{\alpha}}$ in the reduced $m$-dimensional space to a solution $\hat{\mathbf{x}}$ in the original $N$-dimensional space. Moreover, we generally have a prior $\boldsymbol{\alpha}_0$ on $\mathbf{x}$ in $S$ and hence wish to ensure that $\hat{\boldsymbol{\alpha}}$ is reasonably close to $\boldsymbol{\alpha}_0$. Following the standard regularization procedure, the inverse problem can then be cast into the following optimization problem:

$$\hat{\boldsymbol{\alpha}} = \operatorname{argmin}_{\boldsymbol{\alpha} \in \mathcal{S}}\left[\left\|\mathbf{y} - f(d(\boldsymbol{\alpha}))\right\|_2^2 + \gamma\left\|\boldsymbol{\alpha} - \boldsymbol{\alpha}_0\right\|_2^2\right]. \tag{1}$$

Herein, the first term is the data misfit (measuring the distance between the actual measurements $\mathbf{y}$ and those predicted by $\boldsymbol{\alpha} \in \mathcal{S}$), the second term serves as the regularizer (measuring the distance between $\boldsymbol{\alpha}$ and $\boldsymbol{\alpha}_0$), and $\gamma$ is a regularization parameter controlling the influence of the prior $\boldsymbol{\alpha}_0$ on the solution. For simplicity, the regularizer is taken as the square of the $l_2$-norm, however, it could be extended to $\|\boldsymbol{\alpha} - \boldsymbol{\alpha}_0\|_p^q$ ($0 \le p,q < \infty$) for more general cases. We can interpret the transforming operator d as a decoder since it decodes the solution $\hat{\boldsymbol{\alpha}}$ from the reduced space: $\hat{\mathbf{x}} = d(\hat{\boldsymbol{\alpha}})$. In addition, we can view Eq. 1 as the definition of an encoding function e that outputs $\hat{\boldsymbol{\alpha}}$ given $\mathbf{y}$ and $\boldsymbol{\alpha}_0$ as inputs: $\hat{\boldsymbol{\alpha}} = e(\mathbf{y}, \boldsymbol{\alpha}_0)$. Thus, the solution of the original inverse problem can be expressed as $\hat{\mathbf{x}} = d(e(\mathbf{y}, \boldsymbol{\alpha}_0))$.

In practice, a critical challenge in solving the inverse problem lies in representing the prior $\boldsymbol{\alpha}_0$. Besides mathematical language that is routinely used in the context of inverse problems as well as more recently developed priors learned implicitly from quantitative calibration data, human natural language is an indispensable medium for humans to understand and reason about diverse complex phenomena. Hence, we hypothesize that human natural language can flexibly formulate priors that are difficult or impossible to be taken into account by conventional mathematical-model-based regularization methods. This motivates our exploration of semantic regularization in the present paper. Within the realm of semantic regularization, we treat $S$ and $\boldsymbol{\alpha}$ as the semantic manifold and the semantically encoded unknown $\mathbf{x}$, respectively. To represent the semantic prior $\boldsymbol{\alpha}_0$, we leverage a pre-trained LLM: $\boldsymbol{\alpha}_0 = \text{LLM}(\boldsymbol{l})$, where $\boldsymbol{l}$ denotes the semantic description of the prior on the unknown $\mathbf{x}$. Examples of semantic priors are: "it is a piece-wise smooth object", "it is a low-contrast digit-like object", "the subject is raising his arm". As sketched in Fig. 1a, $\boldsymbol{\alpha}$ can be decomposed into two components in the semantic manifold $S$: $\boldsymbol{\alpha} = \boldsymbol{\alpha}_0 + \Delta\boldsymbol{\alpha}$, where the residual term $\Delta\boldsymbol{\alpha}$ represents the deviation from the prior to match the measurements. As a matter of fact, the semantic prior $\boldsymbol{\alpha}_0$ can be automatically estimated from the measurements $\mathbf{y}$, which is helpful when the prior $\boldsymbol{\alpha}_0$ is not otherwise available. Accordingly, the definitions of encoder and decoder can be modified: $(\hat{\boldsymbol{\alpha}}_0, \Delta\hat{\boldsymbol{\alpha}}) = e(\mathbf{y})$ and $\hat{\mathbf{x}} = d(\hat{\boldsymbol{\alpha}}_0, \Delta\hat{\boldsymbol{\alpha}})$, as depicted in Fig. 1b where neural networks implement the encoder and the decoder.

We now elaborate on how to train the decoder and encoder for semantic regularization. Our starting point is a labeled training dataset $\mathcal{D} = \{\mathbf{x}_i, \mathbf{y}_i, l_i; i = 1, 2, \ldots, M\}$ which includes triplets of $M$ realizations of the unknown $\mathbf{x}$, the corresponding measurements $\mathbf{y}$, and the corresponding semantic priors $\boldsymbol{l}$. We use the contrastive learning method to tackle training with this multi-modality dataset given its track record in pairing text with other modalities and its strong zero-shot reasoning capability. We train the encoder and decoder in a supervised manner by minimizing the following loss function:

$$\mathcal{L}(e,d) = \frac{1}{2}\sum_{i=1}^{M}\left(\left\|d(e(\mathbf{y}_i), \boldsymbol{\alpha}_{0,i}) - \mathbf{x}_i\right\|_2^2 + \gamma\left\|\boldsymbol{\alpha}_{0,i} - \text{LLM}(\ell_i)\right\|_2^2 + \left\|\Delta\boldsymbol{\alpha}_i\right\|_2^2\right). \tag{2}$$

The first term in Eq. 2 encourages the encoder-decoder network to explain the observation $\mathbf{y}_i$ by its ground truth $\mathbf{x}_i$ and associated prior

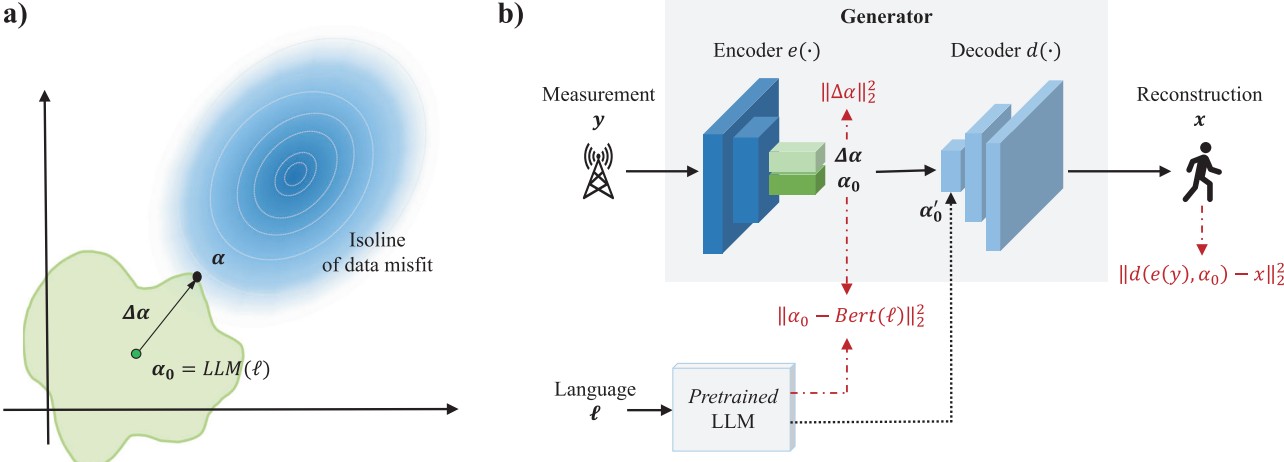

**Fig. 1 | Working principle of semantic regularization based on a pre-trained LLM. a** Illustration of solving Eq. (1) with semantic regularization. The gradient blue graph represents the solution space in the semantic manifold $\mathscr{S}$. Measurements determine the white isolines which represent the data misfit, i.e., the first term on the right-hand side in Eq. (1). The green graph represents the semantic regularizer which is determined mainly by the semantic prior $\boldsymbol{\alpha}_0$, i.e., the second term on the right-hand side in Eq. (1). The optimal solution $\boldsymbol{\alpha}$ is found at the intersection of the data misfit isoline and the semantic regularizer. **b** "Encoder-decoder" neural network architecture for reconstructing the unknown **x** with semantic regularization. The encoder maps the measurements **y** to a pair $(\Delta\boldsymbol{\alpha},\boldsymbol{\alpha}_0)$, where $\boldsymbol{\alpha}_0$ is the semantic prior and $\boldsymbol{\alpha}=\boldsymbol{\alpha}_0 + \Delta\boldsymbol{\alpha}$ is the semantically embedded reconstructed unknown, as illustrated in (**a**). The decoder maps a pair $(\Delta\boldsymbol{\alpha},\boldsymbol{\alpha}_0)$ to the reconstructed unknown in the original space. To train the encoder-decoder network, a multi-step procedure as outlined in Supplementary Note 1 is used. During the first step, the loss function defined in Eq. (2) that is composed of the three terms highlighted in red is minimized. A subsequent GAN-based training step refines the encoder to ensure that it outputs reasonable semantic priors. Once trained, the encoder directly outputs a recommended semantic embedding for a given measurement, which, importantly, can be manually changed into $\boldsymbol{\alpha}_0'$ as required in different contexts explored in this work, before the decoder maps $(\Delta\boldsymbol{\alpha},\boldsymbol{\alpha}_0')$ to a reconstructed unknown in the original space.

$\boldsymbol{\alpha}_{0,i}$, the second term aligns the prior $\boldsymbol{\alpha}_{0,i}$ with its semantic embedding $\ell_i$ via the frozen pretrained LLM, and the last term seeks to ensure that $\Delta\boldsymbol{\alpha}_i$ obeys the standard normal distribution as much as possible in a probabilistic sense.

To ensure that the encoder outputs a reasonable semantic prior $\boldsymbol{\alpha}_0$, we interpret the encoder $(\hat{\boldsymbol{\alpha}}_0, \Delta\hat{\boldsymbol{\alpha}}) = e(\mathbf{y})$ as a part of generator that maps measurements **y** to pairs of semantic priors $\hat{\boldsymbol{\alpha}}_0$ and estimates $\hat{\mathbf{x}} = d(\hat{\boldsymbol{\alpha}}_0, \Delta\hat{\boldsymbol{\alpha}})$ of the unknown **x**. Then, as detailed in Supplementary Note 1, we train a dedicated discriminator that assesses whether the pairs of $\hat{\boldsymbol{\alpha}}_0$ and $\hat{\mathbf{x}}$ are meaningful or not. Finally, we compose a generative adversarial network (GAN)[45] of our generator and discriminator in order to fine-tune the generator. More details about the network are provided in Methods and Supplementary Note 1.

At this stage, it is instructive to elaborate on the relation of our work to previously reported quantitative-data-driven regularization techniques such as those from Refs. 19–22. Our proposed semantic regularization is also data driven; however, it is driven by both quantitative and semantic data. Specifically, it uses triplet training data $\mathcal{D} = \{\mathbf{x}_i, \mathbf{y}_i, \ell_i; i = 1,2,\ldots,M\}$, where $\mathbf{x}_i$ and $\ell_i$ are the quantitative and semantic input data, respectively. Thanks to $\ell_i$, our semantic regularization learns not only the mapping from $\mathbf{x}_i$ to $\mathbf{y}_i$ but also the underlying 'semantic'. In contrast, conventional quantitative-data-driven regularization uses doublet training data $\widetilde{\mathcal{D}} = \{\mathbf{x}_i, \mathbf{y}_i; i = 1,2,\ldots,M\}$. Therefore, conventional quantitative-data-driven regularization is agnostic to semantic information (e.g., originating from human recognition and reasoning). It is apparent that conventional quantitative-data-driven regularization is in fact a special case of our semantic regularization: if $\ell_i$ is not included in $D$, $D$ collapses to $\widetilde{\mathcal{D}}$ and the semantic regularization specializes to the conventional quantitative-data-driven regularization. A detailed comparison between semantic regularization and quantitative-data-only-driven regularization is provided in Supplementary Note 2, where we simply switch off the semantics to consider the quantitative-data-only-driven approach. As detailed in Supplementary Note 2, besides the ability to effectively leverage human reasoning and recognition for regularization, we also observe that the semantic regularization outperforms quantitative-data-only-driven regularization in terms of both generalization capabilities and its robustness to noise. Seemingly, forcing the network during training to represent information in high-level abstracted semantics helps to avoid over-training and being sensitive to noise.

## Numerical results for 2D EM inverse scattering problem

We begin by examining the feasibility of the proposed semantic regularization for a prototypical EM inverse scattering problem[1,5,46–48]. The goal of an EM inverse scattering problem is to determine the scattering properties within a domain of interest (DoI), e.g., the permittivity distribution, based on measurements of scattered fields originating from known excitations. In our numerical study, we consider the 2D setup shown in Fig. 2a. The DoI has a size of 1.28 m × 1.28 m and contains digit-like or/and geometric-shape-like objects with permittivity values in the range of [2, 5]. We use a full-wave solver of Maxwell's equations to generate the data of the scattered fields, as detailed in Methods and Supplementary Note 3. Four transmitters and eight receivers are uniformly placed on a circle of radius 2 m that encloses the DoI, and all 32 possible transmission measurements are determined at the operating frequency of 300 MHz.

Having trained the encoder-decoder network with $M = 60,000$ examples (including the fine-tuning with the discriminator which was trained with 120,000 examples), the solution of the inverse scattering problem consists of two steps, as illustrated in Fig. 2b. First, the encoder maps the measurements (real and imaginary parts are stacked) to the two embeddings $\boldsymbol{\alpha}_0$ and $\Delta\boldsymbol{\alpha}$. Then, the decoder maps the two embeddings $\boldsymbol{\alpha}_0$ and $\Delta\boldsymbol{\alpha}$ to the reconstructed DoI. When semantic prior information about the unknown DoI is known, $\boldsymbol{\alpha}_0$ can be predetermined through the pretrained frozen LLM.

Next, we analyze the impact of $\Delta\boldsymbol{\alpha}$ on the reconstruction. A t-distributed stochastic neighbor embedding (t-SNE[49]) visualization of

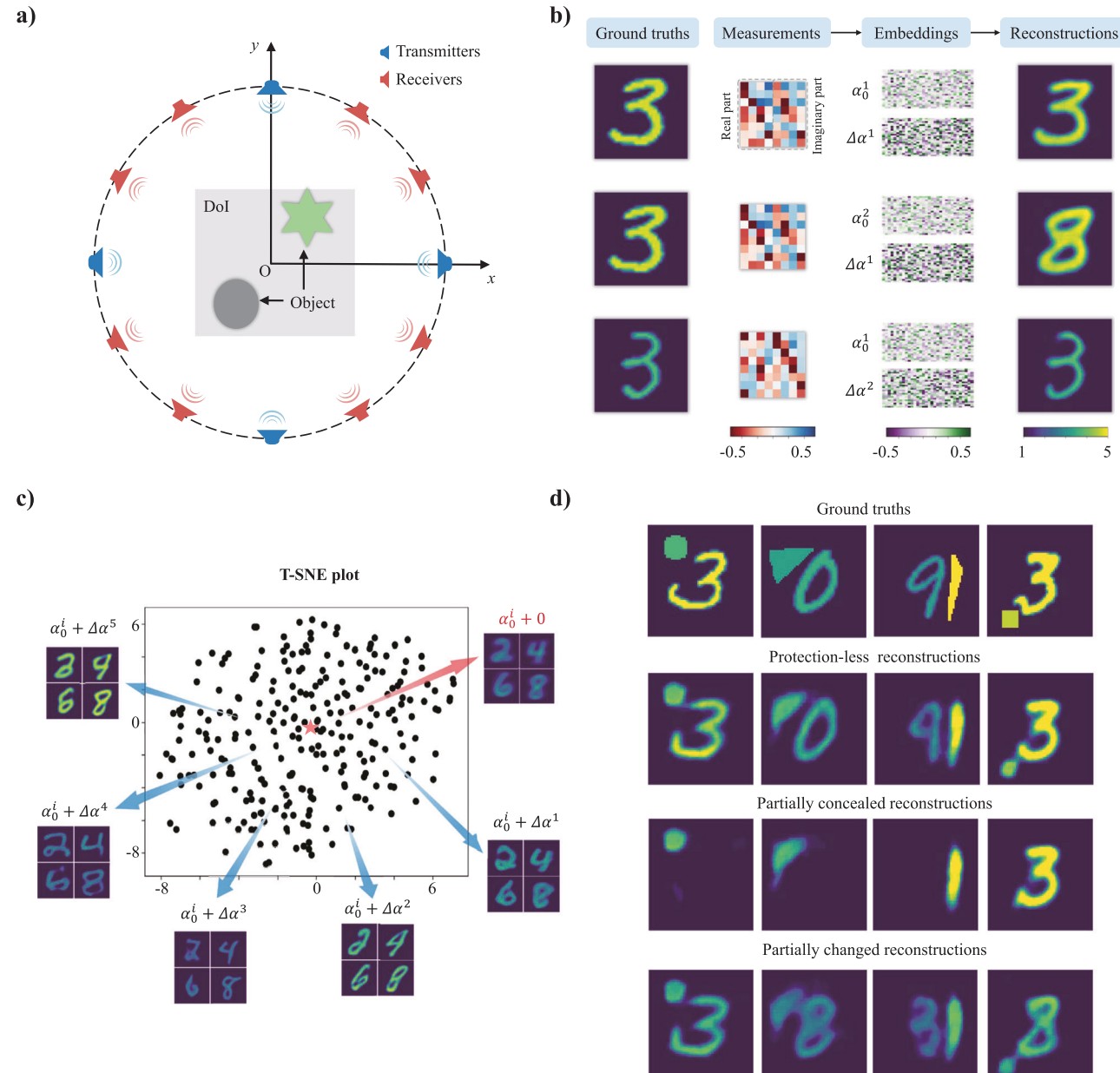

**Fig. 2 | Representative results with semantic regularization for prototypical EM inverse scattering problems. a** Considered configuration of an ill-posed 2D EM inverse scattering problem. **b** Illustration of two-step solution to the inverse scattering problem in the framework of the proposed method showing the measurements, the two embeddings obtained with the encoder (of which $\alpha_0$ is the semantic embedding), and the reconstructed permittivity distributions obtained from the embeddings with the decoder. **c** T-SNE visualization of different $\Delta\alpha$ for a fixed semantic prior. The red star indicates $\Delta\alpha = 0$. The insets show example reconstructions with different $\Delta\alpha$ for a few selected semantic priors. **d** Illustration of the ability to manipulate the reconstruction with the semantic embedding in order to conceal or change a vulnerable object. The ground-truth semantic priors for the ground truth DoIs shown in the first row are: "high-contrast digit-3 and middle-

contrast cycle", "low-contrast digit-0 and low-contrast triangle", "low-contrast digit-9 and high-contrast triangle", "high-contrast digit-3 and middle-contrast square". The second row shows the reconstructions obtained with these semantic priors, i.e., without any protection. To partially conceal the DoI in the reconstruction, the following phrases are integrated into the modified semantic priors in the third row: "conceal digit", "conceal digit", "conceal digit", "conceal shape". To alter the appearance of an object, the following phrases are integrated into the modified semantic priors in the fourth row: "change the digit as middle-contrast digit-3", "change the digit as low-contrast digit-8", "change the digit as low-contrast digit-3 and change the shape as middle-contrast triangle", "change the digit as middle-contrast digit-2 and change the shape as low-contrast square". Details about the MSE evaluation can be found in Supplementary Note 13.

$\Delta\alpha$ for a fixed $\alpha_0$ is displayed in Fig. 2c and the red cross indicates $\Delta\alpha = 0$. The reconstructions $\Delta\alpha = 0$ combined with various realizations of $\alpha_0^i$ yield blurry reconstructions, as seen in the corresponding inset in Fig. 2c. This observation makes sense because using $\Delta\alpha = 0$ is equivalent to averaging the reconstruction for a given $\alpha_0^i$ over many non-zero realizations of $\Delta\alpha$. The remaining insets in Fig. 2c reveal that $\Delta\alpha$ has important effects on the fine-scale details of the reconstruction, i.e., the geometrical style, the physical permittivity values, and so on.

For instance, the reconstructions with $\Delta\alpha_1$ feature round shapes, the reconstructions with $\Delta\alpha_3$ yield notably low permittivity values, the reconstructions with $\Delta\alpha_2$ are lathy and inclined to the right, the reconstructions with $\Delta\alpha_4$ have low permittivity values and wide square shapes, and the reconstructions with $\Delta\alpha_5$ yield bold-font digit shapes. We conclude that $\Delta\alpha$ governs the low-level structural details of the reconstruction whereas the semantic embedding $\alpha_0$ is decisive for the high-level features of the reconstruction.

The observed dependence of the reconstruction on the semantic embedding $\boldsymbol{\alpha}_0$ is particularly valuable in applications with security concerns, e.g., the need to preserve privacy. For instance, by suitably altering the semantic embedding $\boldsymbol{\alpha}_0$, the reconstruction can be manipulated to conceal vulnerable objects/subjects or to change their appearance. In such a scenario, the reconstruction is purposefully manipulated to no longer yield the full objective "truth" about the DoI. Indeed, the reconstruction is constrained into the semantic-defined object space imposed during training and hence only yields (a good approximation of) the objective "truth" if the correct control semantic is provided. However, it is now precisely our goal to obtain a reconstruction in which semantically selected aspects of the DoI are purposefully misrepresented, i.e., "untrue". Representative results are displayed in Fig. 2d, where the DoI contains a composition of a digit-like object and geometric-shape-like object. Compared to the second row in Fig. 2d which shows the reconstruction with the unaltered semantic embedding $\boldsymbol{\alpha}_0$, the ability of a modified semantic embedding to conceal one of the objects or to change its appearance is apparent in the third and fourth rows in Fig. 2d, respectively. For instance, we can conceal the digit in the DoI by changing the semantic embedding, i.e., "It is a composite object: digit-9 and triangle. But, conceal digit." In addition, we can also alter specific parts of the reconstruction by adjusting the control language, e.g., "It is a composite object: low-contrast digit-0 and low-contrast triangle. But, change the digit as low-contrast digit-8." To illustrate the benefits of semantic regularization, for a given DoI the reconstruction results with seven increasingly detailed semantic priors are shown in Fig. 3a. As visually apparent and quantified by the mean square error (MSE), the more detailed the semantic prior is, the higher is the reconstruction quality. To study the influence of the semantic prior on the data misfit and reconstruction quality more systematically, we considered 10,000 pairs of measurements and semantic priors. For a given prior, we quantify how close it is to the ground truth semantic prior $\boldsymbol{\alpha}_0$ by computing $||\boldsymbol{\alpha}_0 - \mathrm{LLM}(\ell)||_2/||\boldsymbol{\alpha}_0||_2$. The data misfit is quantified as $||\mathbf{y} - f(\hat{\mathbf{x}})||_2/||\mathbf{y}||_2$. The reconstruction quality is quantified by the MSE comparing the ground truth $\mathbf{x}$ to the reconstructed $\hat{\mathbf{x}}$. Some reconstruction examples are shown in Fig. 3c. It is apparent that the reconstruction quality rapidly increases with increasing discrepancy between the applied semantic prior and the ground truth semantic prior. Only an accurate semantic prior facilitates the solution of the inverse scattering problem. The general dependence of data misfit and reconstruction quality on the accuracy of the semantic prior is plotted in Fig. 3b (the dots display only 100 from the 10,000 considered pairs of measurements and semantic priors).

We also conducted two additional sets of important numerical experiments regarding the immunity to noise and the generalization capabilities of our method which are detailed in Supplementary Notes 4–7. We found that the proposed method is remarkably robust against unseen noise (i.e., noise that appears during testing that was not present during training): Upon visual inspection the output appears unperturbed for SNR = 20 dB, and for SNR = 5 dB the original basic outline and meaning can still be recognized even though the output is degraded. Moreover, the semantic regularization displays good generalization capabilities. We hypothesize that the semantic regularization in the training process enables the network to more easily grasp the semantically related information, while ignoring the noisy misleading information in the data, resulting in the impact of noise being reduced. In addition, we also provide in Supplementary Note 8 more discussion about the effects of different choices of semantic regularizer settings on the reconstruction quality, for instance, optimizer, learning rate, network architecture, and so on.

### Experimental results for 3D compressive microwave meta-imaging

Having studied the essential properties of semantic regularization with a prototypical numerical inverse scattering problem in the previous section, we now apply semantic regularization experimentally in the context of compressive microwave meta-imaging. The goal of imaging is to determine the scattering properties, e.g., the reflectivity map, of a scene based on measurements of how the scene scatters waves originating from known excitations. To alleviate the transceiver hardware cost, over the last decade the idea of leveraging metamaterial-based hardware for imaging, coined "meta-imaging", has received significant attention. A meta-imager multiplexes scene information across diverse measurement modes offered by the metamaterial's degrees of freedom onto a single (or few) detector(s)[9]. Initially, spectral degrees of freedom were explored[50] but more recently the research focus shifted to configurational degrees of freedom in programmable metamaterials[51], especially because they enable the tailoring of the illuminations to specific types of scenes[52] and even to specific imaging task[9,53–56] and noise types[57]. In meta-imagers, the mapping from scene to measurements is not a one-to-one mapping, requiring a non-trivial computational reconstruction of the scene from the measured data. Often, the dimensionality of the measurements can be remarkably lower than that of the scene because the inherent multiplexing compresses the sparse scene information. Typical compressive imaging problems are ill-posed due to this dimensionality mismatch and the reconstruction relies on sparse regularization. However, in many practical applications, the required transform to obtain a sparse scene representation is unknown.

In this section, we demonstrate experimentally that semantic regularization can simplify the representation of the prior without harsh requirements on knowing the sparse transformation. Moreover, the semantic regularizer has language-controllable properties that can be explored to conceal or alter parts of the reconstructed images. Our experimental measurements are based on a compressive metasurface camera operating around 2.4 GHz (see Methods and Supplementary Note 9 for details). The underlying programmable metasurface is depicted in Fig. 4a. Our goal is to image the posture of two human subjects, Jack (the first author in this article) and Sam (the third author), in our laboratory environment. We train our encoder-decoder network with the same approach as before. Our training dataset includes $M = 25{,}000$ examples; the ground-truth scenes are obtained based on a stereo optical camera, as detailed in Supplementary Note 9. Furthermore, to train the discriminator, 50,000 training examples are created and used. Our training and testing data are collected under the same experimental conditions and include hence the same level of noise. Therefore, the network can optimally adapt to the type and level of noise[57].

Optical images of five representative scenes are displayed in the first row of Fig. 4b. The second row shows the corresponding "protection-less" reconstructions for which the utilized semantic embedding $\boldsymbol{\alpha}_0$ (printed above the reconstructions) is the one automatically proposed by the encoder network based on the microwave measurements. It is apparent that the proposed method not only reconstructs high-fidelity images from the compressive measurements, but it also simultaneously outputs high-level semantic descriptions of the scenes, e.g., "Jack is waving his left hand, and Sam is sitting on the chair". Remarkably, the identities and status of the subjects have been correctly recognized by the algorithm despite the fact that the two subjects have very similar body profiles. Whereas it would be almost impossible to distinguish the two subjects based on optical binarized images or skeletons, the necessary information appears to be encoded in the raw compressive microwave measurements and our semantically regularized compressive metasurface camera is apparently capable of extracting this information from the raw microwave data. Therefore, the proposed method is a promising tool to enable paradigms such as smart homes without infringing the inhabitants' privacy by monitoring the environment with optical cameras. To summarize, the semantically regularized compressive meta-imager is capable of reconstructing the scene in a privacy-

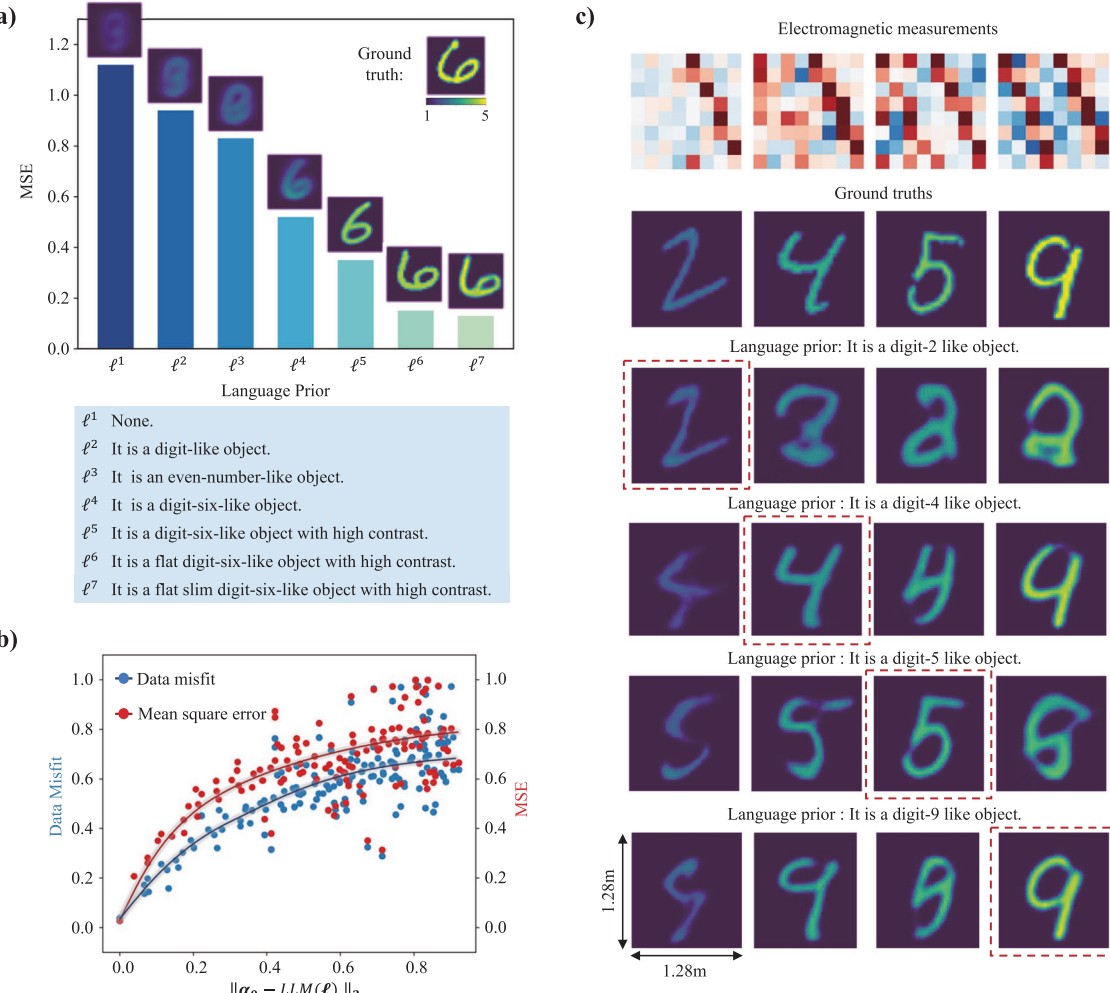

**Fig. 3 | Representative results specifically on the incluence of semantics for prototypical EM inverse scattering problems. a** Impact of increasingly detailed semantic priors on the reconstruction. Each semantic prior is printed in the box and the corresponding MSE values are indicated as bar plot. **b** Assessment of the dependence of data misfit (blue, left vertical axis) and reconstruction quality (red, quantified by MSE, right vertical axis) on how similar the utilized semantic prior

LLM($\ell$) is to the ground-truth semantic prior $\alpha_0$ (horizontal axis). **c** 16 selected examples from the analysis underlying (**b**) are displayed. The top row displays the microwave measurements and the second row shows the corresponding ground-truth permittivity distributions. The subsequent four rows display the reconstructions with different semantic priors. The examples highlighted with red dashed boxes correspond to the use of the ground-truth semantic prior.

preserving manner and simultaneously provides a semantic description of the scene.

Next, we explore the ability of language-controllable imaging to conceal or alter vulnerable parts of the scene. First, we seek to image the posture of one subject while entirely preserving the privacy of the other subject by concealing it via a suitable manipulation of the semantic embedding so that the concealed subject does not appear in the reconstructed image. Corresponding results are displayed in the third row of Fig. 4b. Irrespective of which subject we aim to conceal and what the subjects' postures are, a suitable control command in the manipulated semantic embedding can faithfully identify the subject whose privacy we seek to protect and conceal it, without impacting the imaging of the posture of the subject of interest. It is also possible to manipulate the semantic embedding with a language-based control command that purposefully alters the reconstructed image. For instance, the examples displayed in the fourth row of Fig. 4b show that we can change the postures/actions of the subjects. We can also swap the positions of the two subjects without altering their postures (second column).

Finally, we examine the zero-shot generalization capability of the semantically regularized compressive image reconstruction. We alter the semantic embedding with language-based control commands that

were not seen during the training of the encoder-decoder network. The obtained results are satisfactory, as displayed in Fig. 5a. The use of unseen semantic commands resembling the one proposed by the encoder network (such as "put up his right hand" instead of "raise right arm") yields almost identical reconstructed images. We also investigate the use of three similar semantic commands that are clearly distinct from the semantic embedding proposed by the encoder network, and only the first of the three was included in the training dataset. All three yield very similar reconstructed images in line with the semantically requested modification. The relation between the ground-truth and the unseen new semantic embeddings is visualized on a t-SNE scatter plot in Fig. 5b. The generalization capabilities evidenced in Fig. 5a are very important in sight of the cost of annotating microwave measurements to create sufficient labeled training data—a task that is extremely labor intensive and whose complexity rises significantly as the number of labels increases.

## Experimental results for 4D compressive microwave meta-imaging
We now apply our semantic regularization strategy to a yet more challenging 4D compressive meta-imaging problem in which we seek

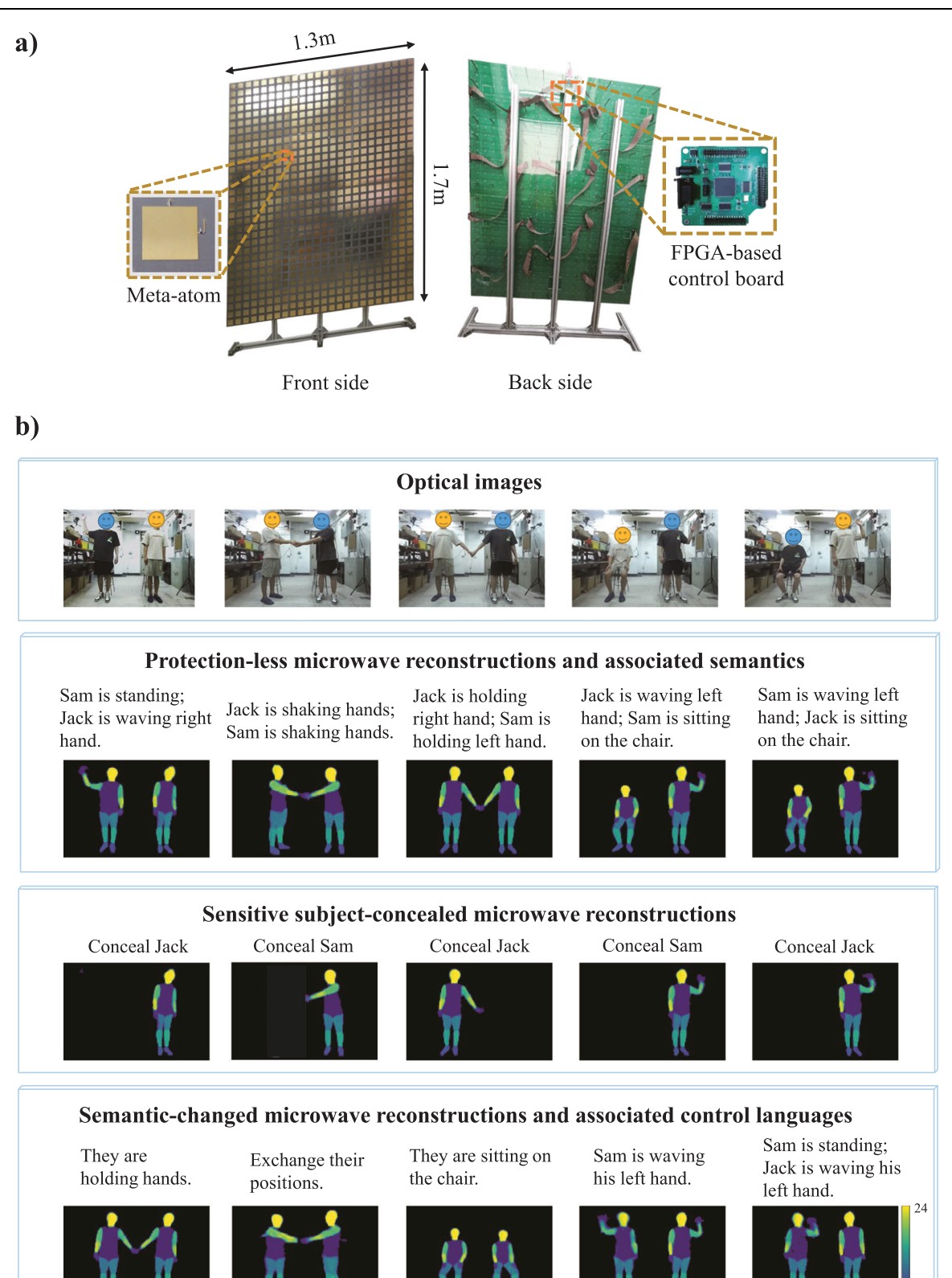

**Fig. 4 | Selected experimental results of semantically regularized 3D compressive microwave meta-imaging. a** Front-view and back-view of the programmable metasurface (1.3 × 1.7 m²). The insets show the programmable meta-atom design and the FPGA-based micro control unit (MCU). **b** Imaging results for a 3 × 2 m² scene containing two freely moving human subjects in a laboratory environment. Optical images of five representative scenes are shown in the first row. The semantic embeddings proposed by the encoder network are used for the reconstructions shown in the second row and indicated there. Different colors represent 24 different body parts. The third and fourth rows show reconstructed images based on altered semantic embeddings that seek to conceal one subject (third row) or alter the subjects' postures/actions (fourth row).

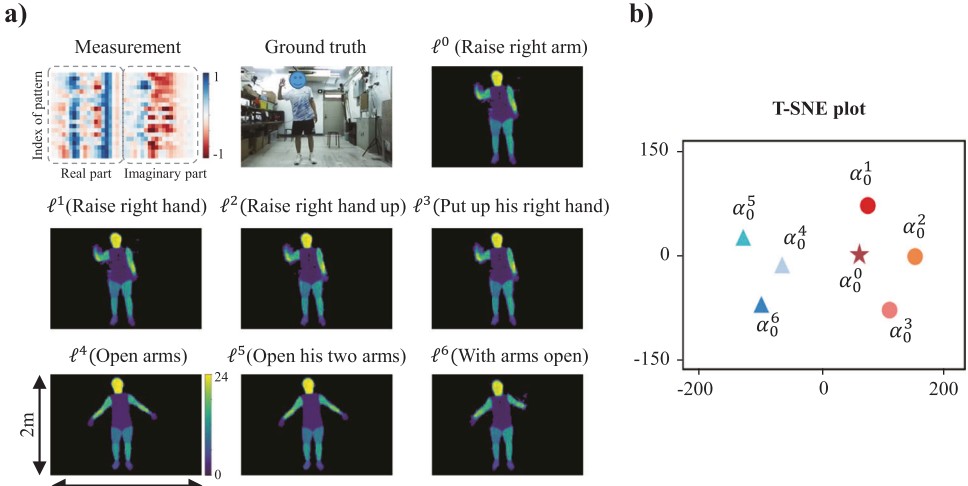

**Fig. 5 | Selected experimental results specifically aimed at the zero-shot generalization capability of the semantically regularized 3D compressive microwave meta-imaging. a** Reconstructions based on semantic embeddings including unseen language-based control commands to manipulate the reconstruction. $\ell^0$ corresponds to the original semantic embedding $\boldsymbol{\alpha}_0^0$ proposed by the encoder network; $\ell^1$-$\ell^3$ correspond to the changed semantic embeddings $\boldsymbol{\alpha}_0^1$-$\boldsymbol{\alpha}_0^3$ which have similar meanings as $\boldsymbol{\alpha}_0^0$ but are not included in the training dataset; $\ell^4$-$\ell^6$ corresponds to the changed semantic embeddings $\boldsymbol{\alpha}_0^4$-$\boldsymbol{\alpha}_0^6$ which are semantically similar but different from $\boldsymbol{\alpha}_0^0$, and only $\boldsymbol{\alpha}_0^4$ is included in the training dataset. **b** T-SNE scatter plot visualizing the resemblance of the semantic embeddings considered in (**a**), i.e., $\boldsymbol{\alpha}_0^0$ − $\boldsymbol{\alpha}_0^6$.

to monitor the spatial-temporal behavior of Sam and Jack in our laboratory environment. The two subjects act continuously and freely in this realistic indoor environment, including actions like hugging, shaking hands, stretching, opening and/or closing drawers, moving objects, etc. In contrast to the 3D imaging problem considered in the previous section, the inputs and outputs are sequences of microwave measurements and 3D images, respectively, in the 4D case. The semantic embedding $\boldsymbol{\alpha}_0$ needs to be consistent with the dynamic action. Correspondingly, the encoder-decoder network needs to be modified such that it is suitable for dealing with these modified inputs and outputs (see details in Supplementary Note 1). We represent the subjects with 19-point 3D skeletons and model the indoor environment with a 3D visual-semantic map such that a given coordinate (e.g., (1.3 m, 1.2 m, 0.8 m)) can be associated with a semantic coordinate (e.g., "at the left side of the chair")[58] (see Fig. 6a and Supplementary Note 10).

Representative results for the case of a single subject (i.e., Jack) are provided in Supplementary Video 1. For the more complex scenarios involving two interacting subjects, representative results are provided in Supplementary Video 2 and corresponding snapshots are also displayed in the second row of Fig. 6b. We distinguish the two subjects by displaying the skeletons in different colors. The high-fidelity reconstruction of our proposed method is apparent and more rigorously quantified by the histogram of the root mean squared errors (RMSEs) of the reconstructed skeletons in Fig. 6c. The RMSE values do not exceed 10 cm. Moreover, the algorithm simultaneously produces accurate high-level semantic recognition results (see also the confusion matrix of semantic recognitions in Supplementary Note 11). Importantly, the semantic descriptions of the scene include details about subjects' identities, actions, and locations, for instance, "Sam and Jack are shaking hands in front of computer". Recall that this detailed information was retrieved without infringing the subject's visual privacy in contrast to what would be possible with optical sensing. Similar to the previous section, we can further protect the subjects' privacy by controlling the semantic component $\boldsymbol{\alpha}_0$. Corresponding reconstruction snapshots results are displayed in the third row in Fig. 6b which are based on purposefully modified semantic embeddings. In particular, for the same set of raw microwave signals, by altering $\boldsymbol{\alpha}_0$ we can change the action or position of any subject in

the reconstruction. The same observations and conclusions as in the previous section follow also in this more challenging 4D imaging context.

## Discussion

To summarize, we introduced and demonstrated the semantic regularization of ill-posed EM inverse problems with the help of pre-trained LLMs. We reported the implementation of regularization that makes use of priors formulated in human natural language rather than mathematical language. On the one hand, this semantic regularization extends the scope of priors that can be considered to those semantically formulated based on human reasoning and recognition for which no simple mathematical formulation exists. On the other hand, semantic regularization enables new forms of privacy protection, for instance, for smart home appliances requiring some level of indoor surveillance. We evidenced that suitable manipulations of the semantic prior can conceal subjects from the reconstruction or alter their appearance in the reconstruction. These capabilities of semantic regularization enable the flexible protection of privacy, e.g., when subjects other than that of interest to the smart home appliance are present. We have implemented the proposed semantic regularization with an encoder-decoder network and applied it to a numerical prototypical 2D inverse scattering problem as well as experimental 3D and 4D compressive imaging problems based on microwave programmable metasurfaces. Our experiments are of direct technological relevance to emerging concepts for smart homes, touchless human-machine interaction, and security screening. Moreover, our work provides new conceptual perspectives on the regularization of inverse problems that can be explored even beyond the considered EM context, as shown in Supplementary Note 14 for an illustrative example in the area of reservoir fluid mechanics.

## Methods
### Training the encoder-decoder network
The architecture of the encoder-decoder network is summarized in Fig. 1b and further detailed in Supplementary Note 1. To train the encoder-decoder network, we proceed in three steps. In the first step, a large-scale labeled triplet dataset $\{\mathbf{x}_i, \mathbf{y}_i, \ell_i; i = 1, 2, \ldots, M\}$ is

**a)**

**c)**

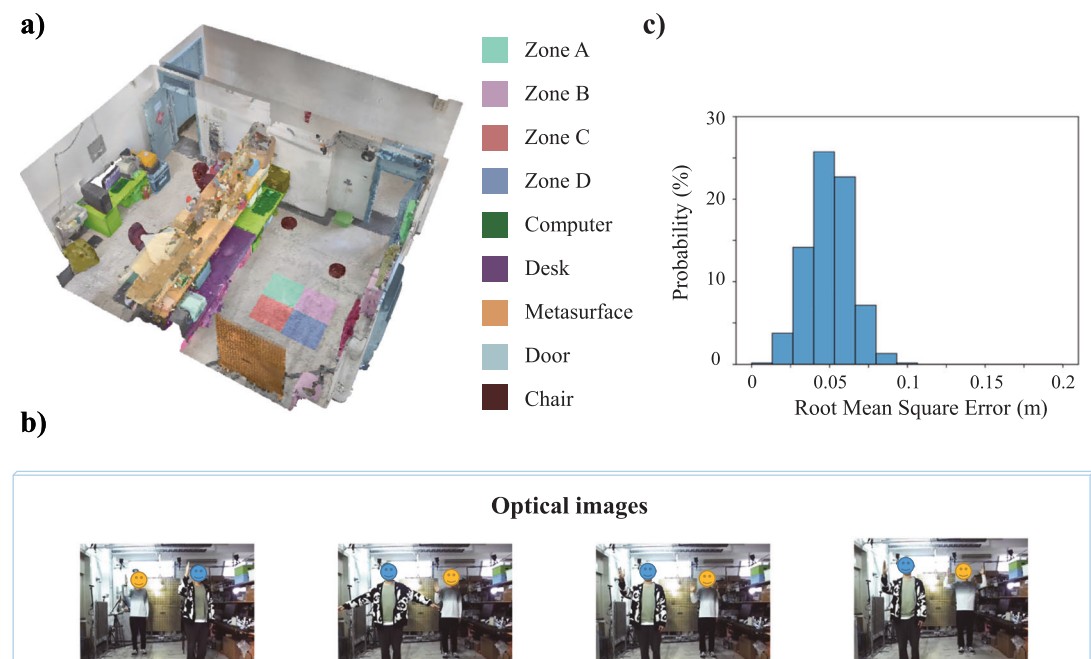

**b)**

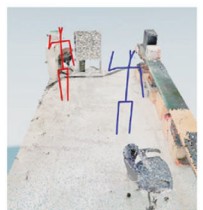
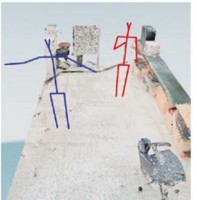
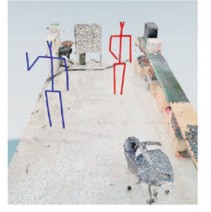
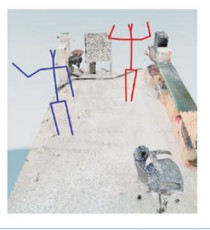

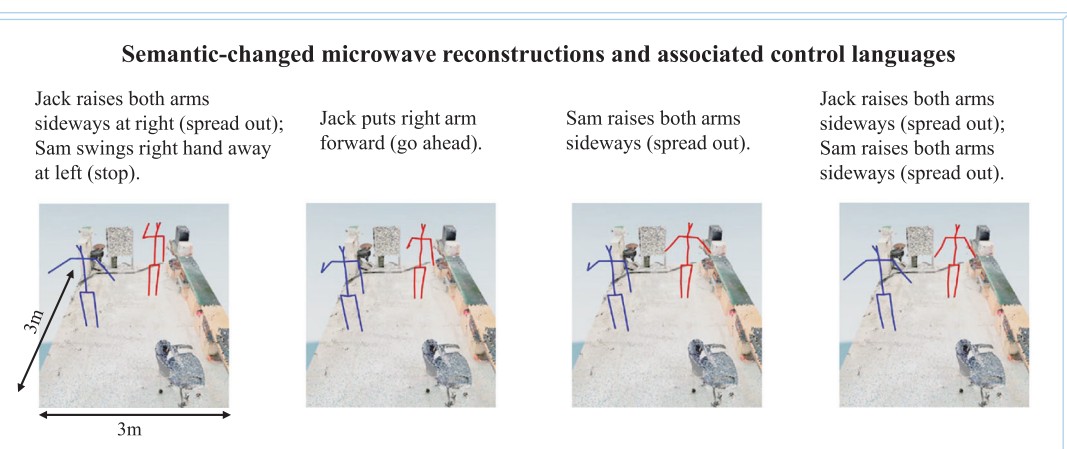

**Fig. 6 | Selected experimental results of semantically regularized 4D compressive microwave meta-imaging. a** 3D visual-semantic map of the indoor environment. **b** Snapshots of 4D imaging results for different laboratory scenes involving two freely acting human subjects. The corresponding optical images are shown in the first row and the reconstructions based on the semantic embedding proposed by the encoder network are shown in the second row. The text in parentheses describes the body language meaning of the action. The third row shows snapshots of semantically altered reconstructions, and the corresponding modified semantic controls are indicated. **c** Histogram of the RMSE distribution comparing the reconstructed 4D skeletons to the corresponding ground truths, based on 10,000 samples for testing.

collected and subsequently used to train the encoder-decoder network. Considering that the ill-posed inverse problem has an infinite number of non-meaningful solutions, i.e., the encoder most likely yields unsuitable semantic embeddings, we integrate the encoder-decoder network as generator together with a discriminator into a GAN in order to fine-tune the semantic embedding produced by the encoder. To this end, in the second step, we collect on-line a large sample of semantic embeddings generated by the encoder network and manually evaluate their meaningfulness, assigning "1" and "0" for correct and incorrect labels, respectively. Then, in the second step, we leverage this new labeled dataset to train the GAN's discriminator that serves as a reward model: it is responsible for scoring the similarity between the semantic embedding output from the GAN's generator (i.e., the encoder-decoder network) and the intended semantics. In the third step, inspired by reinforcement learning[53] or embodied intelligence[23–26], we continuously fine-tune the GAN' generator and discriminator.

### Inverse scattering modeling

We consider a prototypical 2D inverse scattering problem with TM-polarized monochromatic illumination $\mathbf{E}_{in}$. As displayed in Fig. 2a, a nonmagnetic scattering object with relative permittivity distribution $\varepsilon_r(\mathbf{r})$ lies inside the DoI, while $N_{in}$ transmitters and $N_s$ receivers are uniformly distributed along the circle $\Gamma$ surrounding the DoI. The object is successively illuminated by $N_{in}$ transmitters, and the scattered fields $\mathbf{E}_s$ are acquired by $N_s$ receivers for each illumination. The relevant equations read[2,3,38]:

$$\mathbf{E}_s(\mathbf{r}) = j\omega\varepsilon_0 \int_{\text{DoI}} g(\mathbf{r},\mathbf{r}')\chi(\mathbf{r}')\mathbf{E}_t(\mathbf{r}')d\mathbf{r}', \mathbf{r} \in \Gamma. \tag{3}$$

$$\mathbf{E}_t(\mathbf{r}) = \mathbf{E}_{in}(\mathbf{r}) + j\omega\varepsilon_0 \int_{\text{DoI}} g(\mathbf{r},\mathbf{r}')\chi(\mathbf{r}')\mathbf{E}_t(\mathbf{r}')d\mathbf{r}', \mathbf{r} \in \text{DoI}. \tag{4}$$

Herein, $g(\mathbf{r},\mathbf{r}') = -\frac{\omega\mu_0}{4} H_0^{(2)}(\mathbf{k}_0|\mathbf{r} - \mathbf{r}'|)$ is the 2D Green's function in free space, $H_0^{(2)}$ is the second-kind zeroth-order Hankel function, $\chi(\mathbf{r}') = \varepsilon_r(\mathbf{r}') - 1$, and $\omega$ is the angular frequency. The primary purpose of the inverse scattering problem is to reconstruct the distribution of $\chi$ within the DoI from the measurements $\mathbf{E}_s$ along with the corresponding illumination information $\mathbf{E}_{in}$. For our numerical implementation, the DoI is evenly divided into a $N_x \times N_y$ square grid. Further details are provided in Supplementary Note 3. Besides, we here would like to highlight that the proposed semantic regularization strategy could be readily integrated into conventional iterative inverse scattering approaches to remarkably improve the latter's performance. For instance, we developed the semantic-integrated Born iterative method (BIM) to improve the BIM's performance in terms of the reconstruction quality and the convergence behavior, as shown in Supplementary Note 12.

### Compressive metasurface camera

The compressive metasurface camera is a software-defined system enabling high-frame-rate EM sensing. It consists of a programmable metasurface, a low-cost commercial software-defined radio device (Ettus USRP X310), a transmitting antenna, a three-antenna receiver and a host computer. Both the USRP and metasurface communicate with the host computer via an Ethernet connection with a transmission control protocol (TCP); meanwhile, the USRP communicates with the metasurface via I/O series communication. The host computer is responsible for selecting the control patterns and communicates them to the metasurface through the FPGA module; at the same time, it sends a command signal to the USRP in order to synchronize its transmitting and receiving channels.

The programmable metasurface is an ultrathin 2D array composed of meta-atoms with individually controllable reflection properties. In our implementation, the programmable metasurface is composed of $3 \times 3$ identical metasurface panels, and each panel consists of $8 \times 8$ meta-atoms. The size of the designed programmable meta-atom is $54 \times 54$ mm$^2$. Each meta-atom contains one PIN diode which enables the meta-atom to switch between two distinct EM response states. Specifically, the reflection phase response changes by roughly 180° around 2.4 GHz when the PIN diode is switched from OFF (ON) to ON (OFF), while the reflection amplitude remains almost unaltered. The bias voltages of the PIN diodes are controlled by a FPGA-based micro-control-unit with clock of 50 MHz. More details are provided in Supplementary Note 9.

## Data availability

The data that support the findings of this study are available within the supplementary files.

## Code availability

Code that supports the findings of this study is available within the supplementary files.

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

## Acknowledgements

This work was supported by the National Key Research and Development Program of China under Grant Nos. 2023YFB3811502, 2021YFA 1401002. T.J.C. acknowledges the support from the National Natural Science Foundation of China under Grant No. 62288101. L.L. acknowledges the support from State Grid Corporation of China's headquarters technology project "Research on non-contact wireless sensing mechanism and state identification method for distribution network" (5400-202355545A-3-2-ZN).

## Author contributions

L.L. and T.J.C. conceived the idea, and wrote the manuscript. P.d.H. contributed to conceptualization and writing. H.Z. and Y.C. designed and developed the system and conducted the experiments. Z.W. contributed to the experiments. All authors participated in the data analysis and interpretation, and read the manuscript.

## Competing interests

The authors declare no competing interests.

## Additional information

**Peer review information** : *Nature Communications* thanks the anonymous reviewers for their contribution to the peer review of this work. A peer review file is available.

