## [Peer Review File · Nature Communications]

Semantic regularization of electromagnetic inverse problemsReviewer #1 (Remarks to the Author):

In this manuscript, a "semantic" regularization approach is proposed to include a-priori information expressed in natural language in the solution of electromagnetic inverse-scattering problems. The approach not only allows improving the dielectric reconstructions by taking into account text-based information about the scene under test, but is also capable of modifying the obtained solution to conceal some targets or changing its appearance. In other words, a direct human intervention on regularization seems possible. This approach has been validated in two different contexts: the first one, simulated, is related to 2D tomographic quantitative inverse scattering; the second one, experimental, involves compressive meta-imaging in 3D and 4D (i.e., considering time evolution) settings. Results are highly interesting and promising, and the method is quite well presented. I have the following comments:

- In this paper, the semantic regularization is integrated in imaging approaches based on neural networks. It could be interesting to know whether the proposed approach is suitable for the integration with other methods, e.g., iterative inverse-scattering approaches.
- The Authors are invited to clarify the concept of "embeddings", which seems to play a crucial role in exploiting semantic regularization.
- The MSE on the obtained solutions should be presented in all cases when the calculation is possible and meaningful, including the very interesting comparisons with other methods presented as Supplementary Note.
- The possibility of concealing objects and altering the reconstructions is very interesting, but it also poses some serious questions about the "truth" of the result. In other words, it seems both interesting and dangerous at the same time, and it is unclear how, at the end, the data misfit can be minimized, if the reconstructed solution has been significantly altered by the regularization method and it is finally "wrong". Some additional discussions on these points will be greatly appreciated.
- In Supplementary Figure 7, only the ground truth of digit "3" is presented. It would be nice to see the true profiles of all the considered targets.
- There are few typos/inaccuracies. Please proofread the manuscript carefully.

Reviewer #1 (Remarks on code availability):

I saw the code and it seems well organized. However, I was unable to run and verify it, since it needs the related data (not included inside the archive). A good idea could be to distribute a pretrained network with, at least, one complete numerical example.

Reviewer #2 (Remarks to the Author):

The paper presents a novel way to regularize inverse problems via large language models in general and then applies the methodology to the particular inverse problem in electromagnetic imaging. Using large language models in the context of inverse problems is to the best of my knowledge novel. However, in contrast to what has been claimed in the paper, there exists a plethora of data-driven regularizations these days. In fact, there are many review articles and books on this topic all of which have not been mentioned, see e.g. the review article https://www.cambridge.org/core/services/aop-cambridge-core/content/view/CE5B3725869AEAF46E04874115B0AB15/S0962492919000059a.pdf/solving_invers_e_problems_using_datadriven_models.pdf which was written a couple of years ago. The closest paper that comes to my mind is

<https://proceedings.mlr.press/v70/bora17a/bora17a.pdf> and subsequent works. It would be interesting to see how the proposed approach compares to existing ones which are of similar flavors.

The authors introduce inverse problem more widely emphasizing that often solutions are underdetermined. While that is true, most inverse problems also have the property that noise gets amplified during the reconstruction. This aspect has not been mentioned nor analyzed in the present paper.

From reading the paper, I don't think that the results could be reproduced easily. However, this is a common feature seen in the literature more generally.

A big problem for this kind of models is that the reconstruction depends on the solution of a highly nonconvex model, so the optimization algorithm, hyperparameters and starting points affect it. This aspect has not been discussed.

Reviewer #2 (Remarks on code availability):

n/a

Reviewer #3 (Remarks to the Author):

This manuscript and its results are technically sharp and very interesting. My only concern is that it does not deal with a problem of wide, fundamental scientific interest; rather, it concerns with a technological problem: The application of semantically-based machine learning techniques to regularize (and in some sense, to classify, when applied for suitable manipulations of the semantic prior for concealing objects or alter their form in the reconstruction) ill-posed inverse problems involving microwave scattering. I believe that this manuscript would be better suited to a more specialized journal related to machine learning or microwave scattering.

On a minor note, the sentence: "Despite the huge success of mathematically formulated regularizers in mitigating the ill-posedness of inverse problems, these methods struggle or fail to handle prior information originating from human recognition or reasoning because such priors are typically formulated semantically rather than mathematically." is ambiguous. I suggest replacing by "... because the latter priors are typically formulated semantically rather than mathematically."

Reviewer #3 (Remarks on code availability):

n/a

Response Letter

We would like to thank all reviewers for their constructive comments and professional suggestions, which have helped us to improve the quality of our manuscript remarkably. Based on their comments and suggestions, we have carefully and accordingly revised the manuscript (highlighted in yellow). Our detailed point-to-point replies to the issues raised by the reviewers are provided below in blue fonts.

To Reviewer #1

In this manuscript, a "semantic" regularization approach is proposed to include a-priori information expressed in natural language in the solution of electromagnetic inverse-scattering problems. The approach not only allows improving the dielectric reconstructions by taking into account text-based information about the scene under test, but is also capable of modifying the obtained solution to conceal some targets or changing its appearance. In other words, a direct human intervention on regularization seems possible. This approach has been validated in two different contexts: the first one, simulated, is related to 2D tomographic quantitative inverse scattering; the second one, experimental, involves compressive meta-imaging in 3D and 4D (i.e., considering time evolution) settings. Results are highly interesting and promising, and the method is quite well presented. I have the following comments:

Response:

Thanks for your kind support and insightful comments, which will improve remarkably the quality of our manuscript. We have carefully and accordingly revised our manuscript (highlighted in yellow) and our point-to-point replies are provided below in blue. We hope that given our detailed answers and manuscript revisions, the reviewer can approve the revised version of our work for publication in *Nature Communications*.

Comment 1. In this paper, the semantic regularization is integrated in imaging approaches based on neural networks. It could be interesting to know whether the proposed approach is suitable for the integration with other methods, e.g., iterative inverse-scattering approaches.

Response:

We thank the Reviewer for the important question. The proposed semantic regularization approach can indeed be integrated with iterative inverse-scattering approaches. To evidence this, we consider as an illustration the integration of the semantic regularization method with the Born Iterative Method (BIM), a popular inverse-scattering approach. We here briefly discuss it, and more details have been provided in **Supplementary Note 12**.

Formally, the BIM consists of iteratively solving the following linear inverse problem, i.e.,

$$E_s = j\omega\varepsilon_0 G\chi E_t \quad (\text{R1})$$

$$\text{Herein, } E_t = E_{in} + j\omega\varepsilon_0 G\chi E_t. \quad (\text{R2})$$

Note that Eqs. R1 and R2 come from Eqs. 4 and 3, respectively.

Then, substituting the pre-trained semantic decoder $\chi = D(\Delta\alpha, \alpha_0)$ (α_0 is the prior semantic embedding) into Eq. R1, we can arrive at:

$$E_s = j\omega\varepsilon_0GD(\Delta\alpha, \alpha_0)E_t \quad (\text{R3})$$

It is clear that Eqs. R2 and R3 form the foundation of semantic-integrated BIM, as summarized in Table R1.

Table R1. Algorithm of semantic-integrated BIM.

Initializing $\Delta\alpha = 0$;
While (Not arriving at some stopping criterion)
DO
Updating $\Delta\alpha$ by solving $E_s = j\omega\varepsilon_0GD(\Delta\alpha, \alpha_0)E_t$;
Calculating $\chi = D(\Delta\alpha, \alpha_0)$
Updating E_t by solving $E_t = E_{in} + j\omega\varepsilon_0G\chi E_t$
END

To examine the performance of semantic-integrated BIM, a set of numerical experiments have been conducted and corresponding results have been plotted in **Fig. R1**. It can be immediately observed from **Fig. R1** that the semantic-integrated BIM is better than standard BIM in terms of the convergence behavior and the reconstruction quality.

For the purpose of secure reconstruction, as detailed in main text, we here would like to examine the reconstruction quality of semantic-integrated BIM with different control semantics (i.e., α_0), and report a set of numerical results in **Fig. R2**. It can be seen that the semantic-integrated BIM can produce the reconstructions that are matched with specified control semantics, meanwhile, the predicted measurements are well consistent with ground truths.

We added the following comment to the main text: “Besides, we here would like to highlight that the proposed semantic regularization strategy could be readily integrated into conventional iterative inverse scattering approaches to remarkably improve the latter’s performance. For instance, we developed the semantic-integrated Born iterative method (BIM) to improve the BIM’s performance in terms of the reconstruction quality and the convergence behavior, as shown in **Supplementary Note 12**.”

Figure R1 | Comparison between semantic-integrated BIM and conventional BIM. (top row) ground truths, (second row) reconstructions with semantic-integrated BIM, (third row) reconstructions with conventional BIM, (bottom row) convergence behaviors in terms of MSEs vs. iterations.

Figure R2 | Reconstructions of semantic-integrated BIM with different control semantics. (top) ground truth, (middle) reconstructions under different control semantics, (bottom) predictions (red lines) and ground truths (blue lines) of scattering fields corresponding to reconstructions in middle row.

Comment 2. The Authors are invited to clarify the concept of "embeddings", which seems to play a crucial role in exploiting semantic regularization.

Response:

Sorry for this confusion, and thanks for your constructive suggestion.

The term "Embedding" refers to the vector representation obtained by mapping data through a trained neural network into a low-dimensional real-number space. This process allows the neural network to learn meaningful representations of the input data in a more compact form, facilitating various downstream tasks. Here, we directly use the pre-trained LLM "BERT" to transform the text into the corresponding embedding vector, and align the obtained semantic embedding vector from the encoder network under corresponding microwave measurement with it. This allows the computer to better understand the similarities and associations between words, and makes semantic regularization have strong generalization and zero-shot capability.

Specifically, the pre-trained "BERT" transforms each input word into a corresponding embedding vector of length 768. The complete input corresponds to an embedding vector of dimensions $n \times 768$, where n increases with the length of the input sentence, and we can restrict the length n to a fixed value by filling and truncating in practice. The following Fig. R3 illustrates the embeddings corresponding to three different input texts, labeled as "a", "an", and "two". Here, we just take the first dimension and reshape it from a 1×768 vector to a 24×32 matrix. Upon visual inspection, the embeddings for input "a" and "an" are notably more similar than "two", which makes sense and indicates that the pre-trained "BERT" can automatically capture semantic relationships between different input texts, thus achieving strong zero-shot capability.

We made the following modification in our introduction: "LLMs embed closely related concepts near to each other, i.e., they map distinct texts about the same concept to similar low-dimensional real-valued vectors; for instance, the embedding of the word "peach" is closer to that of "fruit" than to that of "knife"."

Figure R3 | Embeddings corresponding to different texts.

Comment 3. The MSE on the obtained solutions should be presented in all cases when the calculation is possible and meaningful, including the very interesting comparisons with other methods presented as Supplementary Note.

Response:

Thanks for your constructive comments!

We have added a supplementary note, i.e., **Supplementary Note 13** (including **16** tables in total), for all necessary MSEs.

We also added the following sentence to the caption of Fig. 3 in the main text: "Details about the MSE evaluation can be found in **Supplementary Note 13**."

Comment 4. The possibility of concealing objects and altering the reconstructions is very interesting, but it also poses some serious questions about the "truth" of the result. In other words, it seems both interesting and dangerous at the same time, and it is unclear how, at the end, the data misfit can be minimized, if the reconstructed solution has been significantly altered by the regularization method and it is finally "wrong". Some additional discussions on these points will be greatly appreciated.

Response:

Thanks for this insightful comment.

First, the semantic is imposed during the training stage with the triple dataset $\mathcal{D} = \{x_i, y_i, \ell_i; i = 1, 2, \dots, M\}$. Then, the reconstruction is constrained into the semantic-defined object space. Therefore, during the testing stage, when the control semantic is changed, the data misfit will be accordingly affected. The results shown in **Fig. R4** indicate that altered reconstructions obtained by controlling semantics lead to increased data misfits of forward scattering field compared to before.

Figure R4 | Reconstructions and corresponding predictions of microwave measurements with ‘changed’ semantics. (top row) ground truths, (second row) reconstructions with original semantics, (third row) reconstructions with changed semantics, (bottom row) predicted microwave fields under original semantics (red lines) and changed semantics (blue lines), where corresponding ground truths of microwave field have also plotted in black lines.

Second, we effectively enforce each reconstruction result to align with its real measurement through supervised paired training during the training process to ensure their correctness. As

a result, the data misfits after changing the semantics are significantly smaller compared to those of other results corresponding to the same semantic but different microwave measurement, as illustrated in **Fig. R5**. In other words, we can still interpret these results as correct matches with the measurements.

We have additionally included the following explanation in our manuscript: “The observed dependence of the reconstruction on the semantic embedding α_0 is particularly valuable in applications with security concerns, e.g., the need to preserve privacy. For instance, by suitably altering the semantic embedding α_0 , the reconstruction can be manipulated to conceal vulnerable objects/subjects or to change their appearance. In such a scenario, the reconstruction is purposefully manipulated to no longer yield the full objective “truth” about the DoI. Indeed, the reconstruction is constrained into the semantic-defined object space imposed during training and hence only yields (a good approximation of) the objective “truth” if the correct control semantic is provided. However, it is now precisely our goal to obtain a reconstruction in which semantically selected aspects of the DoI are purposefully misrepresented.”

Fig. R5 | Statistics of data misfits for two representative sets of reconstructions between their scattered fields and the true microwave measurements with some reconstruction results. Among them, the red triangle corresponds to the reconstruction under the original semantic, the blue star corresponds to the reconstruction after the modified semantic, and the gray dots represent the reconstructions and error under the changed semantic but different microwaves. The ground truths have been depicted in the first two columns of **Fig. R4**.

Comment 5. In Supplementary Figure 7, only the ground truth of digit "3" is presented. It would be nice to see the true profiles of all the considered targets.

Response:

Really sorry for the confusion about Supplementary Figure 7 in the original version. In the original version of Supplementary Figure 7a (b), we considered the reconstructions of '3' ('7') under different noise levels and different semantic constraints.

In order to avoid this confusion, we have revised **Supplementary Figure 7** (now **Supplementary Figure 5.1**), i.e.,

- 1) Reconstructions for five different targets, i.e., '4', '0', '3', '2', '7', under different noise levels in Supplementary Figure 5.1(a).
- 2) Reconstructions of '3' under different noise levels and different semantic constraints in Supplementary Figure 5.1(b).

Comment 6. There are few typos/inaccuracies. Please proofread the manuscript carefully.

Response:

Sorry for the inconvenience and thanks for your helpful suggestions.
Accordingly, we have carefully revised our manuscript.

Comment 7. (Remarks on code availability) I saw the code and it seems well organized. However, I was unable to run and verify it, since it needs the related data (not included inside the archive). A good idea could be to distribute a pretrained network with, at least, one complete numerical example.

Response:

Sorry for the inconvenience and thanks for your helpful suggestions.
In the revised code files, we have added a pre-trained model and corresponding sample dataset for a complete running demo.

To Reviewer #2

The paper presents a novel way to regularize inverse problems via large language models in general and then applies the methodology to the particular inverse problem in electromagnetic imaging.

Response:

Thanks for your kind support and insightful comments, which will improve remarkably the quality of our manuscript. We have carefully and accordingly revised our manuscript (highlighted in yellow) and our point-to-point replies are provided below in blue. We hope that given our detailed answers and manuscript revisions, the reviewer can approve the revised version of our work for publication in *Nature Communications*.

Comment 1. Using large language models in the context of inverse problems is to the best of my knowledge novel. However, in contrast to what has been claimed in the paper, there exists a plethora of data-driven regularizations these days. In fact, there are many review articles and books on this topic all of which have not been mentioned, see e.g. the review article https://www.cambridge.org/core/services/aop-cambridge-core/content/view/CE5B3725869AEAF46E04874115B0AB15/S0962492919000059a.pdf/solving_inverse_problems_using_datadriven_models.pdf which was written a couple of years ago. The closest paper that comes to my mind is <https://proceedings.mlr.press/v70/bora17a/bora17a.pdf> and subsequent works. It would be interesting to see how the proposed approach compares to existing ones which are of similar flavors.

Response:

Thank you very much for very insightful comments and bringing highly relevant works into our attentions. We fully agree that our initial manuscript lacked a contextualization of our work with respect to conventional data-driven regularization. Our revised manuscript provides that contextualization and discusses the proposed references (see Ref. R1-R4), in particular,

[R1] Bora, Ashish, et al. Compressed sensing using generative models. *International conference on machine learning*. (PMLR, 2017).

[R2] Ongie, Gregory, et al. Deep learning techniques for inverse problems in imaging. *IEEE Journal on Selected Areas in Information Theory*. **1.1**, 39-56 (2020).

[R3] Jalal, Ajil, et al. Robust compressed sensing mri with deep generative priors. *Advances in Neural Information Processing Systems*. **34**, 14938-14954 (2021).

[R4] Arridge, Simon, et al. Solving inverse problems using data-driven models. *Acta Numerica*. **28**, 1-174 (2019).

We would like to say that the proposed semantic regularization belongs to data-driven regularization; however, it is remarkably different from existing data-driven regularizations, as discussed below.

First, the semantic regularization is *recognition-aware* because of triple training data $\mathcal{D} = \{x_i, y_i, \ell_i; i = 1, 2, \dots, M\}$. Note that the semantic is another control freedom, which enables to learn not only the mapping relationship between x and y , but also the underlying 'semantic'. In contrast, existing data-driven regularizations are either limited to a certain recognition label or nearly *recognition-agnostic* because of double training data $\mathcal{D} = \{x_i, y_i; i = 1, 2, \dots, M\}$.

Apparently, if the semantics or meaningful semantics are not included in training dataset, the semantic regularization will automatically degrade to conventional data-driven regularization. As a matter of fact, we can observe from **Fig. 2c** that the semantic regularization is powerful in improving the reconstruction quality. Two conclusions can be observed, i.e.,

- i) The more detailed the semantic prior is, the higher is the reconstruction quality
- ii) The semantic regularization enables us to control the semantics as needed to obtain the reconstruction of the same measurement under different semantics using the same network. However, non-semantic data-driven regularizations is lacking in controlling this process, making the interpretability poor.

Second, we also find that the semantic regularization is powerful in improving both the generalizability and the robustness to noise. To see it clearly, we have conducted a set of inverse-scattering numerical experiments, and reported results in **Supplementary Note 2**.

In our implementations, we consider to train two deep artificial neural networks with almost the same architectures and the same size of parameters, but they are trained with or without semantics. Two sets of numerical results have been reported in Fig. R6, in particular,

- i) Results of robustness to different level of noises.
We use the MNIST dataset with the relative permittivity ranging from [2.0, 5.0] for training and testing. Selected reconstruction results with and without semantic regularization for different noise levels have been provided in **Fig. R6(a)** and **Fig. R6(b)**, respectively. It can be observed that the semantic regularization enables the better reconstructions than non-semantic regularization, especially when the signal-to-noise (SNR) level is lower than 10dB.
- ii) Results of generalizability.
In our generalization experiment, we use the MNIST dataset with the relative permittivity ranging from [1.5, 2.0] for training and then directly test with the dataset of geometric shapes with the relative permittivity ranging from [2.0, 3.0]. Selected results with and without semantic regularization have been provided in **Fig. R6(c)**. Now, it is obvious that the semantic regularization has much stronger generalizability than non-semantic regularization, because semantic alignment operations during training enable the network to better capture semantic-related information while disregarding redundant information in data.

Figure R6 | Comparison of reconstruction results between common neural-network-based data-driven regularization method and semantic regularization method. (a-b) Comparison of results for the noise experiment. (c) Comparison of results for generalization experiment.

We revised the following parts of the manuscript to include a contextualization with respect to conventional data-driven regularization:

- **Abstract:** “To date, only prior knowledge that is formulated mathematically (e.g., sparsity of the unknown) or implicitly learned from quantitative data can be taken into account for regularization. Thereby, semantically formulated prior knowledge derived from human reasoning and recognition is excluded.”
- **Introduction:** “More recently, alternative data-driven regularization methods implicitly learned from quantitative calibration data emerged^{19–22}. Despite the huge success of both mathematically-formulated and quantitative-data-driven regularizers in mitigating the ill-posedness of inverse problems, these methods struggle or fail to handle prior information originating from human recognition or reasoning because the latter priors are typically formulated semantically rather than mathematically or quantitatively.”
- **Problem statement:** “In practice, a critical challenge in solving the inverse problem lies in

representing the prior α_0 . Besides mathematical language that is routinely used in the context of inverse problems as well as more recently developed priors learned implicitly from quantitative calibration data, human natural language is an indispensable medium for humans to understand and reason about diverse complex phenomena.”

- **Problem statement:** “At this stage, it is instructive to elaborate on the relation of our work to previously reported quantitative-data-driven regularization techniques such as those from Refs.^{19–22}. Our proposed semantic regularization is also data driven; however, it is driven by both quantitative and semantic data. Specifically, it uses triplet training data $\mathcal{D} = \{x_i, y_i, \ell_i; i = 1, 2, \dots, M\}$, where x_i and ℓ_i are the quantitative and semantic input data, respectively. Thanks to ℓ_i , our semantic regularization learns not only the mapping from x_i to y_i but also the underlying ‘semantic’. In contrast, conventional quantitative-data-driven regularization uses doublet training data $\tilde{\mathcal{D}} = \{x_i, y_i; i = 1, 2, \dots, M\}$. Therefore, conventional quantitative-data-driven regularization is agnostic to semantic information (e.g., originating from human recognition and reasoning). It is apparent that conventional quantitative-data-driven regularization is in fact a special case of our semantic regularization: if ℓ_i is not included in \mathcal{D} , \mathcal{D} collapses to $\tilde{\mathcal{D}}$ and the semantic regularization specializes to the conventional quantitative-data-driven regularization. Besides the ability to effectively leverage human reasoning and recognition for regularization, we also observe that the semantic regularization outperforms quantitative-data-only-driven regularization in terms of both generalization capabilities and its robustness to noise, as detailed in **Supplementary Note 2**. Seemingly, forcing the network during training to represent information in high-level abstracted semantics helps to avoid over-training and being sensitive to noise.”

Comment 2. The authors introduce inverse problem more widely emphasizing that often solutions are underdetermined. While that is true, most inverse problems also have the property that noise gets amplified during the reconstruction. This aspect has not been mentioned nor analyzed in the present paper.

Response:

Thank you so much for constructive comments and suggestions.

First, we would like to highlight that it is important to distinguish between noise that is present in both training and testing data vs. noise that is present only in the testing data but not in the training data.

Our microwave imaging experiments inevitably face some noise, but the noise is the same in both the training and the testing data, such that the network optimally adapts to the type and level noise during training. In most applications, one would ideally collect the training data under the same noise conditions as the testing, as done in our experiments.

Nonetheless, it is also interesting to investigate to what extent the network can cope with noise during testing that was not present during training. We have investigated the impact of different levels of noise on the reconstruction results for the case of our prototypical 2D numerical inverse scattering problem where we can explore the entire range of noise levels because we

start with zero noise and can add arbitrarily large levels of noise to the testing data. The corresponding results can be found in **Supplementary Note 5**. These results prove that the semantic regularization method can obtain good reconstruction results under different semantics within a certain noise range, and the system can recognize the true semantics corresponding to the input measurement. However, when the SNR drops below 20dB, the noise gets markedly amplified and deteriorates significantly the reconstruction.

As shown in **Supplementary Figure 5.1**, the semantic regularization method achieves a very convincing performances under noisy test conditions: almost no degradation for SNR of 20dB, small distortions of the results for SNR of 10dB, and significant distortions for SNR of 5dB. Nonetheless, even for SNR = 5dB, the original basic outline and meaning are maintained and recognizable in the output. We hypothesize that the semantic regularization in the training process enables the network to more easily grasp the semantically related information, while ignoring the noisy misleading information in the data, resulting in the impact of noise being reduced.

We added the following sentences to the main text:

- **Section on 2D numerics:** "We also conducted two additional sets of important numerical experiments regarding the immunity to noise and the generalization capabilities of our method which are detailed in **Supplementary Notes 4-7**. We found that the proposed method is remarkably robust against unseen noise (i.e., noise that appears during testing that was not present during training): Upon visual inspection the output appears unperturbed for SNR=20dB, and for SNR=5dB the original basic outline and meaning can still be recognized even though the output is degraded."
- **Section on 3D experiment:** "Our training and testing data are collected under the same experimental conditions and include hence the same level of noise. Therefore, the network can optimally adapt to the type and level of noise⁵⁷."

Comment 3: From reading the paper, I don't think that the results could be reproduced easily. However, this is a common feature seen in the literature more generally.

A big problem for this kind of models is that the reconstruction depends on the solution of a highly nonconvex model, so the optimization algorithm, hyperparameters and starting points affect it. This aspect has not been discussed.

Response:

Thanks for this insightful comment.

In order to facilitate the reproduction of our results by interested researchers, we have provided some code and data as well as a working demo in the file of supplementary code.

For this purpose, we have also added a supplementary note in the revised version, i.e., **Supplementary Note 8** (including 6 tables), where we evaluate the effects on the MSE-based reconstruction quality of semantic regularization trained with different settings, i.e.,

- 1) Table S8.1: MSE vs. optimizer (SGD, Adam, AdamW, RMSprop and Adagrad)
- 2) Table S8.2: MSE vs. learning rate

- 3) Table S8.3: MSE vs. batch size
- 4) Table S8.4: MSE vs. training epoch
- 5) Table S8.5: MSE vs. initial value
- 6) Table S8.6: MSE vs. network architecture (Note that: The updated supplementary code file contains the code for the specific structures of the models involved.)

To Reviewer #3

This manuscript and its results are technically sharp and very interesting.

Response:

Thanks for your kind support and insightful comments, which will improve remarkably the quality of our manuscript. We have carefully and accordingly revised our manuscript (highlighted in yellow) and our point-to-point replies are provided below in blue.

Comment 1. My only concern is that it does not deal with a problem of wide, fundamental scientific interest; rather, it concerns with a technological problem: The application of semantically-based machine learning techniques to regularize (and in some sense, to classify, when applied for suitable manipulations of the semantic prior for concealing objects or alter their form in the reconstruction) ill-posed inverse problems involving microwave scattering. I believe that this manuscript would be better suited to a more specialized journal related to machine learning or microwave scattering.

Response:

We respectfully disagree with the reviewer's comment. Here, we would like to say that the presented semantic regularization approach can play a fundamental role in generable inverse problems, and can be applicable in various fields besides microwave imaging. To show it, as an illustration example, we have investigated the feasibility of proposed method in the area of reservoir fluid mechanics, and have added **Supplementary Note 14** for this part. Here, a set of results are reported in **Fig.R7**.

It can be observed from our preliminary results that the proposed semantic regularization holds important potential in water resource, subsurface environment, and beyond.

We also added the following sentence to our manuscript: "Moreover, our work provides new conceptual perspectives on the regularization of inverse problems that can be explored even beyond the considered EM context, as shown in **Supplementary Note 14** for an illustrative example in the area of reservoir fluid mechanics."

We hope that given our revised manuscript, the reviewer can recommend the acceptance of our paper in *Nature Communications*.

Figure R7 | Reconstruction results of reservoir fluid mechanics, i.e., history matching, under different observation and semantics. More details can be found in **Supplementary Note 14**.

Comment 2. On a minor note, the sentence: "Despite the huge success of mathematically formulated regularizers in mitigating the ill-posedness of inverse problems, these methods struggle or fail to handle prior information originating from human recognition or reasoning because such priors are typically formulated semantically rather than mathematically." is ambiguous. I suggest replacing by "... because the latter priors are typically formulated semantically rather than mathematically."

Response:

Sorry for this confusion and thanks for helpful suggestions. We have fixed this issue in the revised manuscript (highlighted in yellow).

Reviewer #1 (Remarks to the Author):

I would like to thank the Authors for considering all my comments in the preparation of the revised version of the manuscript. I am satisfied by their replies. I only have a question about Supplementary Figure 5.1 b). Observing the right part of that figure, it seems that varying the SNR there are no changes at all in the reconstructions. I think results should be different, and worse if SNR decreases, in particular. Please check whether the reconstructed images shown in Sup.Fig.5.1b) need corrections.

Reviewer #1 (Remarks on code availability):

The code seems improved with respect to the previous version. However, I cannot fully review it since my computer has no sufficient resources to run it.

Reviewer #2 (Remarks to the Author):

I am happy with most of the authors' response and changes to the manuscript. That being said, I disagree with the authors' response to the question on common data-driven regularization. Their response does not indicate which data-driven method they have chosen. Presumably this is just their own method with the semantics being switched off. There is no single state-of-the-art method but a range of architectures and training scenarios exist, so details are very important. If they wanted to claim superiority over this paradigm (which they currently do), they need sophisticated experiments comparing to the state-of-the-art in at least one relevant training scenario (e.g. supervised learning). However, I would be happy to accept the paper without it, if the authors tone down their claims and not make comparative statements.

Considering the comments by the other reviewers, I agree that this paper is better suited for a specialist journal on imaging.

Reviewer #1 (Remarks to the Author):

I would like to thank the Authors for considering all my comments in the preparation of the revised version of the manuscript. I am satisfied by their replies.

OUR RESPONSE:

We are delighted that the Reviewer is satisfied with our replies.

I only have a question about Supplementary Figure 5.1 b). Observing the right part of that figure, it seems that varying the SNR there are no changes at all in the reconstructions. I think results should be different, and worse if SNR decreases, in particular. Please check whether the reconstructed images shown in Sup.Fig.5.1b) need corrections.

OUR RESPONSE:

We appreciate the Reviewer's attention to Suppl. Fig. 5.1 b). There have indeed been some accidental mistakes in the compilation of this subfigure. We have fixed this issue in the further revised Supplemental Materials. Thank you for noting this.

Reviewer #1 (Remarks on code availability):

The code seems improved with respect to the previous version. However, I cannot fully review it since my computer has no sufficient resources to run it.

OUR RESPONSE:

We are delighted that the Reviewer considers our code to have improved.

Reviewer #2 (Remarks to the Author):

I am happy with most of the authors' response and changes to the manuscript.

OUR RESPONSE:

We are delighted that the Reviewer is satisfied with most of our responses and revisions in the previous round of review.

That being said, I disagree with the authors' response to the question on common data-driven regularization. Their response does not indicate which data-driven method they have chosen. Presumably this is just their own method with the semantics being switched off. There is no single state-of-the-art method but a range of architectures and training scenarios exist, so details are very important. If they wanted to claim superiority over this paradigm (which they currently do), they need sophisticated experiments comparing to the state-of-the-art in at least one relevant training scenario (e.g. supervised learning). However, I would be happy to accept the paper without it, if the authors tone down their claims and not make comparative statements.

OUR RESPONSE:

We thank the reviewer for alerting us to the need to state in our main text what specific data-driven regularization benchmark we considered. The reviewer is right that our purely quantitative-data-driven benchmark method is our own method with the semantics switched off. This was already stated fairly clearly in our Suppl. Note 2: "In the absence of semantics, our method automatically degrades to a standard data-driven regularization approach." and "The kinds of neural networks used as well as the parameter scale are almost identical for both methods." We have now further revised the main text to make sure it is also clearly stated therein that our quantitative-data-driven benchmark is the same method with the semantics switched off:

"A detailed comparison between semantic regularization and quantitative-data-only-driven regularization is provided in **Supplementary Note 2**, where we simply switch off the semantics to consider the quantitative-data-only-driven approach. As detailed in **Supplementary Note 2**, besides the ability to effectively leverage human reasoning and recognition for regularization, we also observe that the semantic regularization outperforms quantitative-data-only-driven regularization in terms of both generalization capabilities and its robustness to noise. Seemingly, forcing the network during training to represent information in high-level abstracted semantics helps to avoid over-training and being sensitive to noise."

Considering the comments by the other reviewers, I agree that this paper is better suited for a specialist journal on imaging.

OUR RESPONSE:

We respectfully disagree with the reviewer's comment that our paper is better suited for a journal specialized on imaging. We are confident that our work introduces a new paradigm of semantic regularization that broadly applies to electromagnetic inverse problems and beyond. As such, we expect our work to constitute a foundational step that opens numerous new research directions in the very broad field of inverse problems. We consider the appeal of our work to extend well beyond communities specialized on imaging and believe that experts with diverse backgrounds ranging from applied mathematics, machine learning and deep learning, electromagnetism, wave physics, metamaterials, wireless communications, fluid dynamics, etc. will find our work to be of interest and a source of inspiration.

Importantly, we have not only proposed the idea but sought to demonstrate it experimentally. Of course, for an experimental demonstration we had to choose a specific context. Hence, we have illustrated our concept experimentally in an imaging context given our background and experimental facilities, and because this example is of direct technological relevance for next-generation wireless networks featuring programmable metasurfaces. However, this choice of context for our experimental demonstration should not be misunderstood as limiting the scope of the proposed concept. Indeed, semantic regularization of inverse problems is a very general concept, and we even provided numerical examples from the strikingly different realm of fluid mechanics.

For all these reasons, we sincerely hope that the reviewer may reconsider their stance and agree that our paper aligns well with the esteemed standards of *Nature Communications*.